# Novel Genetic Markers for Early Detection of Elevated Breast Cancer Risk in Women

**DOI:** 10.3390/ijms20194828

**Published:** 2019-09-28

**Authors:** Bohua Wu, Yunhui Peng, Julia Eggert, Emil Alexov

**Affiliations:** 1Healthcare Genetics, School of Nursing, Clemson University, Clemson, SC 29634, USA; bohua@clemson.edu (B.W.);; 2Computational Biophysics and Bioinformatics, Department of Physics, Clemson University, Clemson, SC 29634, USA; yunhuip@clemson.edu

**Keywords:** MSH2, breast cancer, personalized medicine, early diagnostics, computational modeling, disease-causing mutations, variants

## Abstract

This study suggests that two newly discovered variants in the *MSH2* gene, which codes for a DNA mismatch repair (MMR) protein, can be associated with a high risk of breast cancer. While variants in the MSH2 gene are known to be linked with an elevated cancer risk, the MSH2 gene is not a part of the standard kit for testing patients for elevated breast cancer risk. Here we used the results of genetic testing of women diagnosed with breast cancer, but who did not have variants in *BRCA1* and *BRCA2* genes. Instead, the test identified four variants with unknown significance (VUS) in the *MSH2* gene. Here, we carried in silico analysis to develop a classifier that can distinguish pathogenic from benign mutations in *MSH2* genes taken from ClinVar. The classifier was then used to classify VUS in *MSH2* genes, and two of them, p.Ala272Val and p.Met592Val, were predicted to be pathogenic mutations. These two mutations were found in women with breast cancer who did not have mutations in *BRCA1* and *BRCA2* genes, and thus they are suggested to be considered as new bio-markers for the early detection of elevated breast cancer risk. However, before this is done, an in vitro validation of mutation pathogenicity is needed and, moreover, the presence of these mutations should be demonstrated in a higher number of patients or in families with breast cancer history.

## 1. Introduction

The *MSH2* gene is a DNA mismatch repair (MMR) gene and its protein product has a crucial function for detecting and repairing DNA errors [NM_000251.1]. The MSH2 protein [NP_000242.1] binds two other proteins, MSH6 or MSH3, to form a protein complex. The complex’s function is to identify the errors in DNA during DNA replication and recombination [1]. The MSH2/MSH6 complex recognizes base to base mismatch and small insertion/deletion errors, while the MSH2/MSH3 complex initiates repair of longer insertions and deletions [1,2]. Both complexes recruit downstream MMR proteins like MLH1/PSM2, or the MLH1/MLH2 complex to complete the repair process [1,3].

Mutations in the *MSH2* gene were reported to be associated with the Lynch syndrome (LS), also known as hereditary nonpolyposis colorectal cancer (HNPPC) [4]. Mutations in the *MSH2* gene were also observed in patients affected with adenocarcinoma of the colon and endometrial carcinoma [5,6,7]. Other researchers reported that LS is different from common colon cancer (with sporadic mutations) that has an earlier onset age [8]. Moreover, *MSH2* mutations carriers were found to have a cumulative risk of colorectal cancer, independent of gender, based on research of 121 families with LS [9]. More than half of the LS cases were attributed to *MSH2* mutations [10,11].

In addition to its association with colorectal cancer, *MSH2* mutations are linked with hematological malignancy, gastrointestinal, urinary tract, ovary and other types of cancers as well [12,13,14,15,16]. Even more, some researchers demonstrated that *MSH2* mutation carriers have an increased risk of breast cancer (BC) with or without a LS family history [15,17,18,19]. A novel *MSH2* mutation, NM_000251.2:n.2781G>T, was identified in a breast cancer patient with a family history of BC, ovarian cancer and some other Lynch syndrome (LS)-associated cancers [20]. The patient did not have mutations in commonly tested breast cancer genes, *BRCA1, BRCA2, CHEK1* or *RAD51C*, but the same mutation in the *MSH2* gene was also identified in other family members [20]. This finding prompted the need to complete genetic testing of the *MSH2* gene, when *BRCA1* and *BRCA2* are without mutations, but multiple family members have BC history [20].

The *MSH2* gene is located on chromosome 2p21-p16.3 with 16 exons [NM_000251.1]. The MSH2 protein contains 934 amino acids and can be grouped into five domains (Figure 1): Mismatch binding, connector, lever, clamp, and ATPase domains. Domain 1 is the mismatch binding domain, containing amino acids 1 to 124; domain 2 is the connector, comprised of amino acids from 125 to 297; domain 3 is the lever, consisting of two segments, which are the residues from 300 to 456 and residues 554 to 619; domain 4 is the clamp domain, containing the residues from 457 to 553, which are located between the two segments of the lever domain; and domain 5 is the ATPase domain, consisting of the residues from 620 to 934 [21,22]. The MSH2 protein is expressed in the nucleus, nucleoplasm, and membrane, and is widespread within lymph nodes, testis, colon and 24 other tissues.

The majority of pathogenic mutations in the *MSH2* gene has been shown to alter DNA-protein interactions, protein-protein interaction, ATPase activity, allosteric signaling between DNA and ATP binding locations, and protein stability [21,22]. Some mutations, for example: p.Val161Asp, p.Cys333Tyr, p.His639Tyr and p.Cys697Phe, were found to be deleterious, affecting allosteric signaling, and the deficiency was confirmed in vitro [21,23]. p.Ala636Pro was predicted to be pathogenic and to affect the stability of the protein structure. This result was consistent with the in vitro experiment indicating that the structure of *MSH2* is damaged [21,23]. Mutations in the *MSH2* gene may cause structural defects of the MSH2 protein, which typically alter the protein structure or changes binding and interactions [21]. This may explain the prevalence of the low expression level of the MSH2 protein in tumor tissues.

There are 40 pathogenic missense mutations in *MSH2* gene reported in ClinVar, a public archive of human gene variants associated with diseases [24,25]. However, numerous variants in *MSH2* are still listed in the unknown significance category. More specifically, 1238 missense mutations in the *MSH2* are not classified. Furthermore, understanding the pathogenicity effect of *MSH2* variants would enhance the knowledge of drug resistance and sensitivity of tumor therapy [26].

For the purpose of our investigation, variants in the *MSH2* gene were retrieved from ClinVar [24], Exome Aggregation Consortium (EXAC), the 1000 Genomes Browser [27] and NHLBI-GO Exome Sequencing Project online databases [28]. They were grouped into “pathogenic and benign”, and their effects of various MSH2 protein characteristics (as structural stability, evolutionary conservation score, etc.) were modeled. The results were used to generate a classifier that discriminates pathogenic from benign mutations. The classifier was then used to classify four mutations identified in patients with BC and two of them, p.Ala272Val and p.Met592Val, are predicted to be pathogenic mutations and associated with elevated BC risk.

## 2. Results and Discussion

To identify what physicochemical characteristics and thermodynamics parameters are essential to discriminate pathogenic versus benign mutations, we systematically examined 22 features (Table 1). Those 22 physicochemical characteristics were investigated for their plausible linkage with disease. Below, we outline the outcome for each of these characteristics and parameters.

### 2.1. Mapping All Missense Mutations onto the 3D Structure of MSH2 Protein

The dataset contained 41 non-pathogenic mutations, 34 pathogenic mutations, and 4 VUS. Mapped onto the 3D structure of MSH2 protein, one can see that most of the non-pathogenic mutations were located on domain 1 (mismatch binding domain), domain 2 (connector domain 3.1 lever) and domain 5 (ATPase binding domain) (Figure 1). In contrast, no pathogenic mutations were located on domain 1. The largest number of mutations, both pathogenic and non-pathogenic mutations, were found on domain 5. The VUS were located in domains 1, 2, 3.2 and 4 (Figure 1). Thus, with the exception of the observation that there were no pathogenic mutations in domain 1, the rest of the MSH2 protein structure harbored mutations of all types.

### 2.2. Evolutionary Conservation Score (ECS)

The evolutionary conservation score (ECS) represents the conservation of a given residue type at the same location of the same protein in various species. A higher ECS indicates that the residue type was conserved and any mutation was expected to be unwanted. The results indicate that 156 residues have an ECS over 0.9, including 13 pathogenic mutations and four non-pathogenic mutations (Figure 2). Most of the highly conserved residues were located on domains 2, 3.1 and 5. A total of 12 out of 34 pathogenic mutations were located on domain 3.1. Fifteen locations with pathogenetic mutations (44%) had an ECS over 0.9 and only five locations with pathogenic mutations (15%) had an ECS lower than 0.8 (Figure 3A). In contrast, only four locations with non-pathogenic mutations (10%) had an ECS higher than 0.9 (Figure 3A). The results showed that most of the locations with pathogenic mutations were located at highly conserved sites, while the locations with non-pathogenic mutations were not.

### 2.3. Folding and Binding Free Energy Changes

Folding free energy change presents the change of protein stability due to mutation(s), and the binding free energy change presents the impact of mutation(s) on protein-protein interactions. To further assess the differences between pathogenic mutations and non-pathogenic mutations, the changes of folding and binding free energy were calculated with various approaches: DUET [29], I-mutant 3.0 [30], mCSM [31], SAAFEC [32], SDM [33], PoPMuSiC [34] BeAtMuSiC [35], MutaBind [36], and SAAMBE [37].

The prediction results indicate that most non-pathogenic mutations had no effect on protein stability, but pathogenic mutations destabilized the protein structure (Figure 3B). There were 14 non-pathogenic mutations resulting in a folding energy change within 0.5 kcal/mol (0.5 ≥ ΔΔG ≥ −0.5 kcal/mol). Such small changes (within 0.5 kcal/mol) were usually considered as non-significant. Comparing changes of ΔΔG due to pathogenic and non-pathogenic mutations, the ΔΔG changes due to pathogenic mutations were larger. None of the pathogenic mutations resulted in ΔΔG between −0.5 kcal/mol and 0.5 kcal/mol, and around 80% of the ΔΔG of pathogenic mutations were greater than 1 kcal/mol or below −1 kcal/mol (Figure 3B). Any large change of the folding energy, making the protein less or more stable than the wild-type, was considered to be greatly affecting protein function [38,39].

The binding energy change (ΔΔΔG) results did not show obvious differences between pathogenic and non-pathogenic mutations (Appendix A). The prediction for non-pathogenic mutations showed most of the binding free energy changes (ΔΔΔG) upon non-pathogenic mutations were between −0.5 kcal/mol and 0.5 kcal/mol with a small standard deviation. However, the prediction results made from different webservers frequently showed controversial results. All pathogenic mutations were not located on the protein-protein interface, which may explain why the binding free energy changes failed to discriminate pathogenic from non-pathogenic mutations.

### 2.4. Molecular Dynamics (MD) Simulation

All of the mutants described above, and the wild-type protein were subjected to extensive MD simulations. The simulation results showed that pathogenic and non-pathogenic mutations had different protein conformation dynamics. The root mean square deviations (RMSDs) for the Cα of each mutant and wild-type protein were calculated (Figure 3C). Most RMSD values for non-pathogenic mutation protein were close to the value of wild-type proteins (Figure 3C). In contrast, the majority of pathogenic type proteins had an extremely large average RMSD. The larger RMSD indicated lower stability of the corresponding protein.

The root mean square fluctuations (RMSF) values of Cα were computed to determine if the wild-type or the mutated residue affected the dynamic conformation of the corresponding location of the protein. The RMSF value of each residue for wild-type and mutant type proteins were calculated in order to examine the flexibility of the protein structure at that position in the wild-type and mutant proteins. We observed that RSMF values of positions of mutations calculated on the corresponding mutant protein did not discriminate between pathogenic and non-pathogenic mutations. However, if the RMSFs were calculated with the wild-type protein, we found that 90% of pathogenic mutation locations had a RMSF value between 1−2 Å, while non-pathogenic mutation locations have the opposite trend (Figure 3D). The low RMSF value implicated that the pathogenic mutation was located in a more rigid region where the residue was less flexible and had limited movement during the simulation than the non-pathogenic mutations.

A mutation may alter the flexibility and induce conformational change to nearby residues. In order to examine the regional RMSF change, the cumulative RMSF of 11 neighborhood residues were calculated by summing up the RMSF value of the position of the variant (using either wild-type or the corresponding mutant structure), and five neighborhood residues before and after the variant (with respect to the amino acid sequence). However, we did not observe any trend indicating that this quantity could discriminate between pathogenic and non-pathogenic mutations (Appendix A).

### 2.5. B-Factors

B-factors provided in the corresponding Protein Data Bank (PDB) file are indicators of conformational flexibility of individual atoms. Thus, we attempted to see if B-factors at the corresponding location of the wild-type protein could potentially discriminate the pathogenic and non-pathogenic mutations. The results were not inclusive. Indeed, Appendix A shows the plot of RMSFs versus B-factors and while one appreciates that non-pathogenic mutations were much more scattered compared with pathogenic, the signal was not strong. However, one observed that the B-factor of most pathogenic mutation positions fell between the interval of 80–85 and data points were clustered. Thus, the pathogenic mutation locations tended to have a smaller RMSF, and B-factor values compared with non-pathogenic sites.

### 2.6. Relative Solvent Accessible Surface Area (rSASA) and B-Factors

The relative solvent accessible surface area (rSASA) for both wild-type and mutant type residue were calculated with Equation (3). The results show that 20 of 34 wild-type residues in the pathogenic mutation group had a rSASA under 0.1. This indicates that most of the pathogenic mutations were in buried positions in the protein structure.

The distribution of B-factors and rSASA values were examined for wild-type residues (Appendix A). Pathogenic mutations had a smaller B-factor and rSASA than non-pathogenic mutations. This finding infers that the pathogenic mutations in MSH2 involved residues that were more rigid and buried than those of non-pathogenic mutations.

### 2.7. Protein Distance (PD)

The PD (Equation (1)) was tested as a discriminator between pathogenic and non-pathogenic mutations. The results are shown in Appendix A. The PD was small for most mutations in the non-pathogenic group, while PD was large for the pathogenic group (Appendix A).

### 2.8. Hydrogen Bonds

The H-bonds were calculated with Equation (4) for both mutant and the wild-type residues, and the changes of the total H-bonds were counted. The results did not show significant differences between pathogenic and non-pathogenic groups (Appendix A).

### 2.9. Receiver Operating Characteristics (ROC)

The main goal of this investigation was to identify physicochemical characteristics and thermodynamics parameters, which can discriminate between pathogenic or non-pathogenic mutations and be used as an input for a classifier (Table 1). To optimize the performance of the prediction model, these 22 physicochemical characteristics and thermodynamics parameters were analyzed with ROC (Table 1). The area under the curve (AUC) of ROC indicates whether a predictor is better than random. In developing a classifier, one wants to optimize the performance while reducing the number of input parameters (to avoid over-fitting). Thus, we selected a particular threshold for AUC to reduce the number of input parameters. The AUC results for folding free energy change, evolutionary conservation score, the average RMSD of the protein, and the RMSF of the wild-type residue position were higher than 0.75, which made us select them as good classifiers.

### 2.10. Selecting the Best Predicting Protocol

Folding free energy change, evolutionary conservation score, the average RMSD of the protein and the RMSF of the wild-type residue were selected as classifier in K-nearest neighbors (KNN) and support vector machine (SVM) methods (see method section). The dataset, including 34 pathogenic missense mutations and 41 non-pathogenic missense mutations, was randomly allocated into a training dataset (52 mutations) and a testing dataset (23 mutations), and then subjected to KNN and SVM classification. KNN analysis was performed with or without RMSD and RMSF, and we tested different K values to find the optimal value of K. SVM analysis was also performed with or without RMSD/RMSF. Linear, polynomial, radial and sigmoid kernel were tested in SVM with the optimal cost and gamma value as well. The classification shows better performance without RMSD and the accuracy is 100% when the K value is six or eight in the KNN method. The prediction accuracy was lower in the SVM method. Finally, the KNN classification method using folding free energy change, evolutionary conservation score and the RMSF of wild-type residues was selected as the best predictor.

### 2.11. Classification of VUS Using KNN Method

The KNN method was utilized to classify the four VUS from clinical data (Table 2). It predicted that VUS p.Ala272Val and p.Met592Val were pathogenic, while p.Tyr43Cys and p.Asn547Ser were non-pathogenic. Furthermore, the classification was assessed via the RMSF, B-factors and rSASA of wild-type residues positions (Appendix A). The results showed that the corresponding quantities at positions 272 and 592 were into the area that most pathogenic mutations were located, while the quantities associated with positions 43 and 547 were away from those areas.

The prediction results were compared with Polyphen and SIFT predictions. Polyphen and SIFT predicted p.Ala272Val to be pathogenic, which is consistent with our results, and both of them gave contradictory results on p.Tyr43Cys and p.Met592Val, which were classified as non-pathogenic and pathogenic by our model with the KNN method. However, their predictions were considered not to be reliable since we observed underestimated and overestimated predictions, applying Polyphen/SIFT on the known pathogenic/non-pathogenic mutations (Appendix A).

### 2.12. Clinical Features of VUS

In this work, four VUS were investigated, which were detected in newly diagnosed breast cancer patients from the Clemson University (CU) and Greenwood Genetic Center (GGC) research project. To examine whether those mutations have an impact on breast cancer risk, here we analyzed the outcome of the predictions with corresponding clinical data (Table 2).

Four of 186 patients were identified as *MSH2* missense mutation carriers that met the requirements of this study. Genetic test results, mutation type and bio-markers for each carrier were retrieved. To protect the patients’ privacy, four carriers were coded as B1, B2, B3 and B4 for this study. The physiochemical features of the VUS were analyzed along with clinical characteristics of the carrier with VUS (Appendix A).

The B2 patient carried the p.Ala272Val mutation and the B4 patient carried the p.Met592Val mutation. Both mutations were predicted as pathogenic by our model (note that while p.Ala272Val mutation is reported to reduce protein function [40], this mutation is still reported as a mutation with unknown clinical significance based on variant classification guideline by the American College of Medical Genetics and Genomics(ACMG)). Furthermore, previously this mutation was mentioned in a patient with breast cancer, but the patient had p.R3128X in BRCA2 as well [41]). Referring to the clinical information of those two carries, the B2 patient had a high grade, stage I breast cancer, with the size of the tumor smaller than 1 cm. The triple negative bio-markers of estrogen (ER), progesterone (PR) and Her2 complicated the diagnoses. This mutation carrier had a breast cancer family history, but no genetic testing information was included. The p.Met592Val mutation was also predicted to be deleterious. This patient, B4, with the p.Met592Val mutation had intermediate, stage IIA breast cancer with a tumor size larger than 1cm. The patient was negative for both PR and Her2 markers but the estrogen marker was positive. As mentioned, both MSH2 mutation carriers, B2 and B4 were diagnosed with breast cancer, while they did not have mutations in *BRCA1,2* genes.

The B1 patient carried the p.Tyr43Cys mutation, which was predicted to be non-pathogenic. This patient carrier was diagnosed with an intermediate, bilateral, stage III breast cancer with positive estrogen and progesterone tumor markers. This B1 mutation carrier also has a family history of BC and GI cancer. When the genetic test result was checked on this carrier, the B1 carrier also had a *BRCA2* mutation (NM_000059.3(BRCA2):c.4936_4939delGAAA) which was previously confirmed, by many studies, to be an associated breast cancer risk [42].

The B3 patient carried the p.Asn547Ser mutation, which was predicted to be non-pathogenic. However, this patient had a mutation in the *BRCA2* gene (NM_000059.3(BRCA2):c.8791A>G). The carrier was diagnosed with an intermediate, stage IIA breast cancer with a tumor size less than 1 cm. The ER and Her2 tumor markers were both negative for this carrier.

Thus, the pathogenic mutations in *MSH2* gene suggested in this study were found in breast cancer patients without variants in *BRCA1,2* genes, while the other two patients carrying non-pathogenic mutations in MSH2 gene had *BRCA2* gene variants.

## 3. Materials and Methods

### 3.1. Selection of Pathogenic MSH2 Mutations

The pathogenic mutations investigated in this work were selected from the ClinVar database [24,25]. The search was queried using the search term “MSH2” in the ClinVar database [NM_000251.1] (Figure 3). The results were further refined by using the string “missense mutations”. The results consisted of the classifications: Benign (19), likely benign (67), uncertain significance (1238), likely pathogenic (53), pathogenic (40), and conflicting reports of pathogenicity (31) [24,43]. Among them, we selected only mutations that were classified as pathogenic. Furthermore, the mutations with controversial interpretation, or resulting from a single submission were removed from the list, and this resulted in a total of 34 pathogenic mutations.

### 3.2. Selection of Benign MSH2 Mutations

Non-pathogenic (benign) mutations were selected from two sources (Figure 4). One set was obtained from ClinVar Database [24]. The other set was obtained utilizing EXAC [44], the 1000 Genomes Browser [27] and NHLBI-GO Exome Sequencing Project [28]. The explanation of how these two data sets were obtained is provided below.

The search term “MSH2” was used to query the ClinVar database. This resulted in the identification of 19 variants classified as “benign” and 67 variants classified as “likely benign.” Excluding those variants with controversial interpretations, the final set of non-pathogenic variants resulted in 20 missense mutations from the ClinVar Database.

The second set of mutations was obtained by using the “*MSH2* gene” query within the EXAC browser. The results were further refined by using the “missense mutations” string. In total, 429 missense mutations in the *MSH2* gene were identified. To further identify non-pathogenic mutations delivered from the EXAC database, the 1000 Genomes Project and ESP (NHLBI-GO Exome Sequencing Project) database sets were used to narrow the selections. EXAC database includes the whole genome sequencing data from 60,706 unrelated individuals. Individuals participating in the 1000 Genomes Project were all in healthy conditions, while the ESP database includes cases of genes contributing to heart, lung and blood disorders. Thus, the following two selection criteria were applied on the mutations taken from ClinVar: (a) If a mutation was not detected in the 1000 Genomes Projects, the 1000 Genomes Browser, Ensembl GRCh38 (http://useast.ensembl.org/Homo_sapiens/Info/Index) and (b) if a mutation was present in the ESP database, in both cases the mutation was removed from the list. Thus, 26 mutations were identified as non-pathogenic (benign) mutations.

Based on the two sources outlined above, the final set of non-pathogenic mutations was comprised of 41 non-pathogenic missense mutations. Out of them, 20 non-pathogenic mutations were found from the ClinVar and 26 non-pathogenic mutations were found from the 1000 genomes project. Only 5 non-pathogenic mutations were common in ClinVar database and the 1000 genomes project dataset.

### 3.3. MSH2 Missense Variants of Uncertain Significance

Uncertain significance missense variants were identified from the joint research project of the Clemson University School of Nursing and Greenwood Genetic Center entitled “Collaborative Oncology Testing: A New Model for South Carolina” (Figure 4). Institutional review board (IRB) approval for the study was obtained from the Bon Secours Richmond Health System IRB according to 45 CFR 46.111 (see Appendix A: IRB approval documentation).

After being diagnosed with breast cancer between 1 July 2014 and 30 June 2017, a total of 218 patients were recruited from a community cancer center in the upstate of South Carolina. A total of 186 patients provided peripheral blood samples and they were examined by next-generation sequencing [44]. Variants were confirmed with Sanger sequencing and classified according to the American College of Medical Genetics (ACMG) Standards and Guidelines for the Interpretation of Sequence Variants [45]. The ACMG guidelines classifies variants into 5 categories: (1) Benign, (2) likely benign, (3) pathogenic, (4) likely pathogenic, or (5) variant of unknown significance (VUS).

Seven VUS in the *MSH2* gene were found in seven individuals in this group of patients. Since our investigation highly relies on the structure, one of these seven VUS, the VUS p.Thr934Met, is located in a region (residues 855–934) where the 3D structure of MSH2 protein is missing, thus it was excluded from the study. Another two VUSs, c.93C > T and c.2785C > T were also excluded since one was a synonymous mutation and one was a stop codon (note that the patient with a stop codon had a mutation in BRCA2 gene as well, which is an additional reason not to be included in the analysis). Overall, four VUS missense mutations, p.Tyr43Cys, p.Ala272Val, p.Asn547Ser and p.Met592Val were selected for this study.

### 3.4. Preparation of 3D Structure of MSH2

The crystal structure of the MSH2 protein [NP_000242.1] was obtained from the Protein Data Bank (PDB) [46]. The PDB file (ID: 2O8B) contains chain A, chain B and a short DNA segment, but misses some residues in the structure [21]. Chain A is the MSH2 protein and Chain B is the MSH6 protein. The profix module from the Jackal package was used to rebuild those missing heavy atoms and short loops in the MSH2 and MSH6 protein (Figure 5) [47].

### 3.5. Property Distance (PD)

Peng et al. used property distance (PD) to quantitatively describe the physical-chemical property differences between wild-type and mutant residues [48]. The residue’s physical–chemical properties were assessed by a property vector, which included two elements: Hydrophobicity and charge. The experimentally determined hydrophobicity scale was utilized as a reference for the hydrophobicity of residues [49,50,51]. Regarding the change, residues Arginine (Arg) and Lysine (Lys) carry a +1 charge, aspartic acid (Asp) and glutamic acid (Glu) carry a −1 charge, and all other residues are considered neutral with a charge of 0. Thus, the property distance between wild-type and mutant residue was calculated using the following equation [48]: (1)$$\mathrm{PD}\left(x,y\right)=\;\sqrt{{\left(H\left(x\right)-H\left(y\right)\right)}^{2}+{\left(Q\left(x\right)-Q\left(y\right)\right)}^{2}}$$
where x and y represent wild-type and mutant residue, respectively; H and Q represent hydrophobicity and charge for the corresponding residue, respectively.

### 3.6. Evolutionary Conservation Score (ECS) Calculations

The MSH2 sequence of 73 different species were downloaded from UniProt [52]. Multiple sequence alignment was performed with the T-Coffee webserver [52,53]. The ECS of each residue of human MSH2 sequence was calculated with the following equation: (2)$$\mathrm{ECS}\left(i\right)=\frac{N\left(i\right)\mathrm{identical}}{N\left(i\right)\mathrm{total}}$$
where “i” is the sequence position of the residue based on human MSH2 sequence. The N(i)_identical_ depicts the number of identical residues for the corresponding location “i” in the multiple sequence alignment and N(i)_total_ = 73.

### 3.7. Folding Free Energy Change (ΔΔG) and Binding Free Energy Change (ΔΔΔG)

Several webservers were applied to predict the effect of mutations on protein folding and binding free energies. The webservers used to analyze folding free energy change in this work include DUET [29], I-mutant 3.0 [30], mCSM [31], SAAFEC [32], SDM [33], and PoPMuSiC [34].

The webservers used for determining the binding free energy change in this study included BeAtMuSiC [35], mCSM [31], MutaBind [36], and SAAMBE [37].

### 3.8. Relative Solvent Accessible Surface Area (rSASA) and Hydrogen Bond (H-bond) Number Calculations

The rSASA and H-bonds were calculated using Visual Molecular Dynamics (VMD) [54]. The rSASA for residues were calculated using the following formula:(3)$$\mathrm{rSASA}\left(i\right)=\frac{\mathrm{SASA}\left(i\right)}{\mathrm{SASA}\left(i\right)\max}$$
where SASA(i) is the solvent accessible surface area calculated for the corresponding residue “i” in the protein, and SASA(i)max is the maximum possible solvent accessible surface area for the corresponding residue “i” alone.

H-bonds were calculated by the total hydrogen bond number of bonds, being acceptor or donor bonds [55].
N_total H-bond(i)_ = N_acceptor(i)_ + N_donor(i)_(4)
where N_total H-bond(i)_ is the total hydrogen bonds that residue “i” is involved, which is split into hydrogen bonds that residue “i” serves as an acceptor (N_acceptor(i)_) and a donor (N_donor(i)_). The cutoff distance was 3.5 A and the cutoff angle was 60 degree. The hydrogen bonds were calculated using energy minimized structure.

### 3.9. B-Factor of the Alpha Carbonate of the Corresponding Residue

The X-ray crystallographic B-factor is a parameter which indicates the flexibility of an atom. The B-factor values of the wild-type alpha carbonate (Cα) atoms were extracted from the MSH2 protein crystal structure (PDB ID:2O8B).

### 3.10. Molecular Dynamic (MD) Simulations

Once the missing residues were rebuilt by the profix package, 79 mutant structures (p.Ala2Thr, p.Pro5Gln, p.Pro5Leu, p.Thr8Met, p.Phe23Leu, p.Met26Leu, p.Thr32Ser, p.Tyr43Cys, p.Arg96His, p.Val102Ile, p.Arg106Lys, p.Asn127Ser, p.Met141Val, p.Val161Asp, p.G162Arg, p.Val163Gly, p.Val163Asp, p.Gly164Arg, p.Gly164Glu, p.Asp167His, p.Ile169Val, p.Arg171Lys, p.Leu173Arg, p.Leu187Arg, p.Leu187Pro, p.Glu198Gly, p.Cys199Arg, p.Val200Asp, p.Lys228Glu, p.Ser268Leu, p.Ala272Val, p.Val273Ile, p.Leu310Pro, p.Gly322Asp, p.Cys333Tyr, p.G338Glu, p.Pro349Leu, p.Pro349Arg, p.Arg359Ser, p.Leu390Phe, p.Gln419Lys, p.Leu440Pro, p.Arg444Leu, p.Metet453Lys, p.His466Arg, p.Ser479Asn, p.Asn547Ser, p.Ser554Cys, p.Ser554Gly, p.Ser554Thr, p.Thr564Ala, p.Ile577Thr, p.Gly587Arg, p.Met592Val, p.Asp597Ala, p.Pro622Leu, p.Gln629Arg, p.Ala636Pro, p.His639Tyr, p.Ala640Ser, p.Val642Ile, p.Val655Ile, p.Tyr656Cys, p.Glu658Gly, p.Gly669Val, p.G683Arg, p.Met688Ile, p.Met688Arg, p.G692Arg, p.Pro696Leu, p.Cys697Arg, p.Cys697Phe, p.Ile704Thr, p.Gly751Arg, p.Gly759Glu, p.His785Pro, p.Glu809Lys, p.Ala834Thr, p.Lys845Glu) were generated from the wild-type MSH2–MSH6 protein complex structure using the VMD 1.9.3 mutator package [54].

Molecular dynamic (MD) simulation was performed using NAMD2.11 with a Charmm36 force field [56]. The protein structure first underwent energy minimization for 10,000 steps for all simulations in order to relax possible overlaps. Generalized Born implicit solvent (GBIS) was applied in the simulations and the time step was set to 1 fs. The temperature in the simulation was set to 300 K. Frame was outputted every 2500 steps in the simulation and the DCD files were subjected to VMD for further analysis. Root mean square deviation (RMSD) of the Cα and root mean square fluctuations (RMSF) of a residue were also calculated using VMD 1.9.3.

### 3.11. K-Nearest Neighbors (KNN) and Support Vector Machine (SVM) Classifications

K-nearest neighbors (KNN) and support vector machine (SVM) algorithms were trained and tested on a total of 75 missense mutations, out of which 34 pathogenic and 41 non-pathogenic. This dataset was randomly split into a training dataset (52 mutations) and a testing dataset (23 mutations). The KNN classification was performed using R studio and optimal K values were tested to obtain the best performance for the data. The SVM classifier with different kernel was also utilized in R. The training dataset and test dataset were the same as those used in the KNN classification.

## 4. Conclusions

We analyzed the effect of 22 physicochemical characteristics and thermodynamics parameters on wild-type and 75 mutant MSH2 proteins in order to identify the optimal classifiers, which could discriminate pathogenic from non-pathogenic mutations. We found that the wild-type residues corresponding to pathogenic mutations were highly conserved. In contrast, non-pathogenic mutations were located on the positions that had a low evolutionary conservation score. Folding free energy change showed that only non-pathogenic mutations had a value from −0.5 kcal/mol to 0.5 kcal/mol. A total of 90% of wild-type residues where pathogenic mutations were found had a RMSF value between 1–2 Å, while most wild-type residues, where non-pathogenic mutations were found, had a larger RMSF value. These findings indicate that pathogenic *MSH2* mutations were in more rigid regions and those mutations affect protein stability, flexibility and conformational dynamics.

In this work, the evolutionary conservation score, folding free energy change and RMSF of wild-type residues were able to discriminate pathogenic from non-pathogenic mutations, thus those three characteristics were applied in the KNN classification model. VUS p.Ala272Val and p.Met592Val were predicted as pathogenic, and VUS p.Tyr43Cys and p.Asn547Ser were predicted as non-pathogenic. The predictions were consistent with clinical observations.

This finding strongly suggests that p.Ala272Val and p.Met592Val in *MSH2* genes should be considered for inclusion in genetic testing profiles for breast cancer. Especially for patients without *BRCA1* or *BRCA2* mutations. However, for such an inclusion, an in vitro validation of mutation pathogenicity is needed and, moreover, the presence of these mutations should be demonstrated in a higher number of patients or in families with BC history. If our findings are confirmed, then this will expand the spectrum of bio-markers and will provide better options for early detection, intervention and diagnosis.

## Figures and Tables

**Figure 1 ijms-20-04828-f001:**
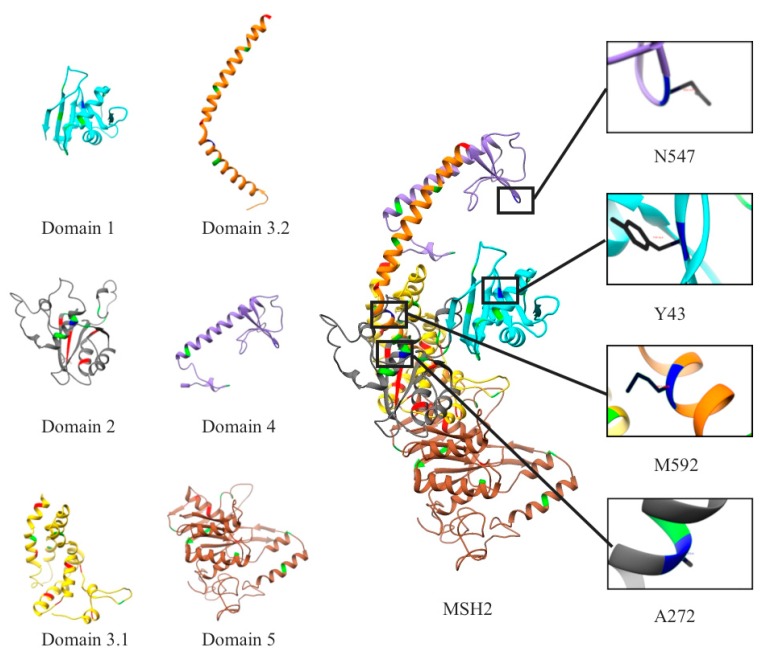
The 3D Structure of the MSH2 Protein. Each domain is depicted in a different color: Domain 1(cyan) mismatch binding domain, domain 2 (dark gray) connector, domain 3.1 (yellow) levers, domain 3.1 (orange) levers, domain 4 (purple) clamps, and domain 5 (sienna) ATPase domain. Differentiation among the mutations is depicted with pathogenic in red, non-pathogenic in green and unknown significance (VUS) in blue.

**Figure 2 ijms-20-04828-f002:**
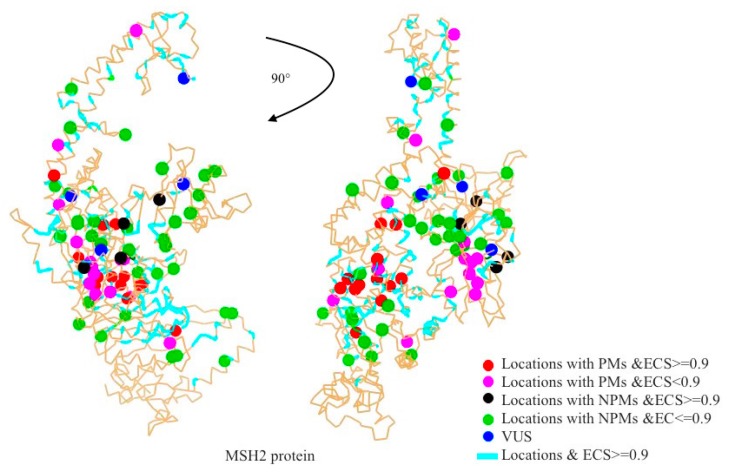
Visualization of highly conserved residues and mutations mapping onto the MSH2 protein structure. PMs: Pathogenic mutations, NPMs: Non-pathogenic mutations, VUS: Variants of unknown significance, and ECS: Evolutionary conservation score.

**Figure 3 ijms-20-04828-f003:**
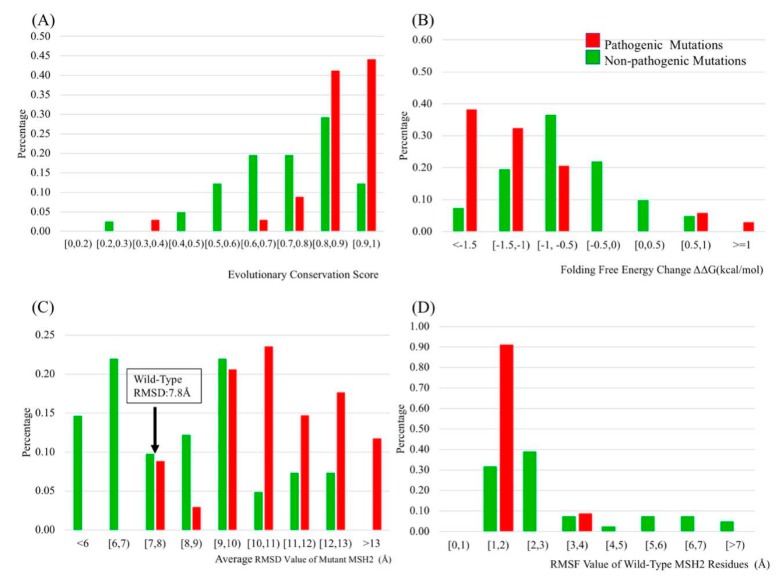
The distribution of evolutionary conservation score, folding free energy change upon the mutations, the average RMSD value of the MSH2 Cα, and the RMSF value of the MSH2 wild-type residues. (**A**) There were around 2/3 of the non-pathogenic mutations with an EC score below 80%; in contrast, more than 80% of pathogenic mutations have an evolutionary conservation score (ECS) higher than 80%. (**B**) This figure demonstrates that around 30% of non-pathogenic mutations had a ∆∆G between −0.5 kcal/mol and 0.5 kcal/mol. However, most of the pathogenic mutations had a larger ∆∆G comparing to non-pathogenic mutations. (**C**) The average RMSD value of wild-type MSH2 protein was 7.8 Å. Some non-pathogenic mutants had a RMSD value lower than the wild-type, but pathogenic MSH2 mutants usually have a much higher value than the wild-type. (**D**) The RMSF value of wild-type residues were calculated. Residues with a corresponding pathogenic mutation were labeled as red, and residues with a corresponding non-pathogenic mutation were labeled as green. More than 90% of wild-type residues on pathogenic positions had a RMSF between 1 to 2 Å, but most of wild-type residues on non-pathogenic positions have a larger RMSF value.

**Figure 4 ijms-20-04828-f004:**
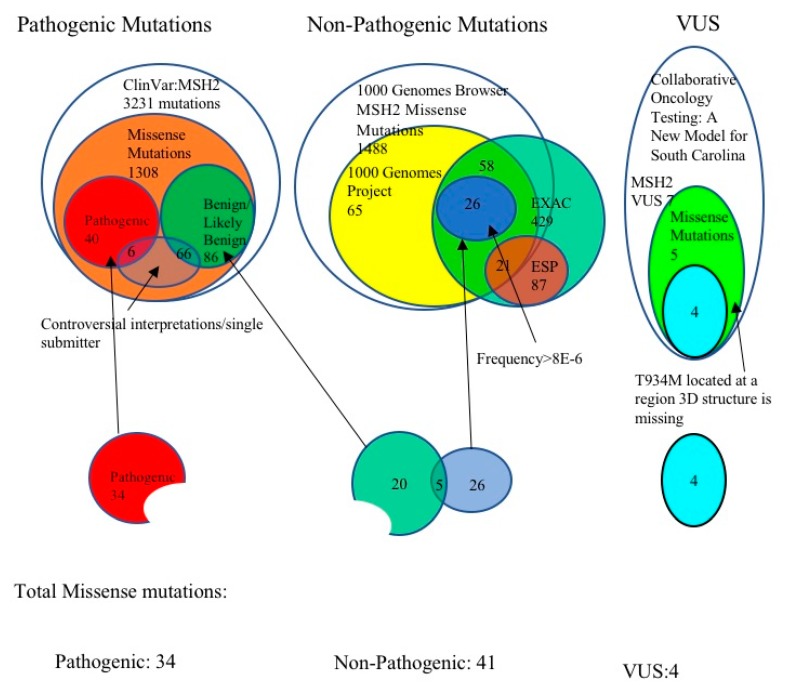
The selection procedures for pathogenic, non-pathogenic and VUS *MSH2* Mutations. Pathogenic mutations were retrieved from the ClinVar database, non-pathogenic mutations were retrieved from the 1000 Genomes project with EXAC database and ESP database; and VUS mutations were from the research project, “Collaborative Oncology Testing: A New Model for South Carolina”.

**Figure 5 ijms-20-04828-f005:**
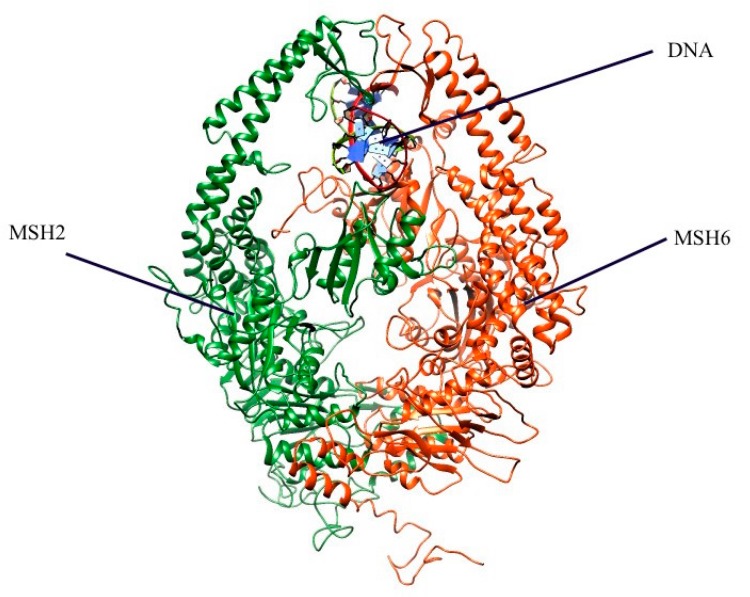
Fixed 3-D structure of the MSH2–MSH6 Complex. The depictions include the proteins for MSH2 in green and MSH6 in red. The DNA sequence is in blue.

**Table 1 ijms-20-04828-t001:** Area under the curve (AUC) value of receiver operating characteristics (ROC). Twenty-two physicochemical characteristics and thermodynamics parameters being investigated. Protein structures being analyzed: Wild-type 1; non-pathogenic mutations 41; pathogenic mutations 34.

Classifiers	AUC Value	Classifiers	AUC Value
Folding free energy change	0.77	H-bond numbers of mutant residues	0.54
Binding free energy change	0.53	H-bond numbers of wild-type residues	0.52
Evolutionary conservation score (ECS)	0.81	H-bond number change upon mutations	0.52
Average root mean square deviations (RMSD) of all protein structures	0.80	B-factors of wild-type residues	0.69
RMSD change upon mutations	0.65	Relative solvent accessible surface area (rSASA) of mutation residues	0.69
Root mean square fluctuations (RMSF) of mutant residue	0.69	rSASA of wild-type residues	0.72
RMSF of wild-type residue	0.80	rSASA change upon mutations	0.50
RMSF change upon mutations	0.73	Protein Distance	0.70
Cumulative RMSF of mutant	0.57	Residue size change upon mutations	0.70
Cumulative RMSF of wild-type	0.61	Residue charge change upon mutations	0.54
Cumulative RMSF change upon mutation	0.55	Polarity change upon mutations	0.70

**Table 2 ijms-20-04828-t002:** Physicochemical characteristics and thermodynamic parameters of VUS with prediction results and clinical data. Folding ∆∆G: Folding free energy change; ECS: Evolutionary conservation score; RMSF_WT: RMSF of wild-type residue; BC: Breast cancer; GI: Gastro-intestinal cancer.

MSH2 VUS Mutations	p.Tyr43Cys	p.Ala272Val	p.Asn547Ser	p.Met592Val
Folding ∆∆G	−1.228	−0.788	−0.26	−1.286
ECS	0.767	0.877	0.863	0.575
RMSF_WT	4.559	1.966	5.725	1.876
Prediction	Non-pathogenic	Pathogenic	Non-pathogenic	Pathogenic
BC Mutations	BRCA2: c.4936_4939delGAAA	N/A	BRCA2: c.8791A>G	N/A
Clinical feature	Intermediate bilateral stage III size>1	High grade stage I BC size<1cm	Intermediate, stage II A size<1cm	Intermediate stage II A
Family History of BC/GI	Yes/Yes	Yes/No	Yes/No	Yes/No

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
