# Peer review of "Novel Genetic Markers for Early Detection of Elevated Breast Cancer Risk in Women"

_ijms, 2019, doi:10.3390/ijms20194828_

Round 1

Reviewer 1 Report

Wu et al. presented the manuscript indicated that two newly discovered variants in MSH2 gene, which codes for a 12 DNA mismatch repair (MMR) protein, can be associated with high risk of breast cancer. The conclusion suggests that two mutations can serve as new biomarkers for early detection of elevated for breast cancer. The study provides some insight to for the early diagnostics breast cancer. These studies were carried out carefully and have completed the writing describes the techniques and findings clearly.

Author Response

Thank you for your time.

Reviewer 2 Report

The manuscript describes the possible association of two newly discovered MSH2 gene variants with BC risk. Authors used the pathogenic and non-pathogenic MSH2 mutations retrieved from several database to develop a classifier considering several physicochemical parameters. The classifier was then used to predict the pathogenicity of 4 VUS mutations identified in their cohort of BC patients. Overall, the manuscript is interesting and clearly written. However, some issues should be addressed. 

Abstract

1. Page 1, line 14: Did the Authors mean “kit” instead of “kid”?

2. Authors should correct MHS2 with MSH2. Authors should check this issue also throughout the text.

3. I think that the last statement “...serve as new biomarker…” is too strong and overestimates the findings. To support Authors conclusions, an in vitro validation of mutation pathogenicity is needed and, moreover, the presence of these mutations should be demonstrated in a higher number of patients or in families with BC history. Conclusions need to be rewritten.

Results and Discussion

4. Page 11, line 286: Authors stated that the p.Ala272Val variant is reported in ClinVar as a mutation with unknown clinical significance, but I found that it was classified as "Benign" on Oct 18, 2018. Authors should clarify this point.

5. Page 11, line 292: Authors should check the mutation reported for patient B4. Some lines above, Authors wrote that patient B4 carries the p.Met592Val mutation and not the p.Ala272Val. 

6. Page 11, line 296: Here and throughout the text Authors should change “BRACA” with “BRCA”.

Materials and Methods

7. Since its damaging effect on protein, it could be interesting a comment on the stop codon mutation that Authors found but that they excluded. Is it present in a patient without mutation in BRCA genes? Is there a family history of BC? What about the clinical features of this patient?

Conclusions

8. Page 18, lines 481-482: Similarly to Abstract, the sentence should be rewritten. Authors should also point out that the impact of the two mutations should be confirmed in vitro and in a larger cohort of patients or in relatives with BC history. Otherwise, since Authors investigated only MSH2 gene, it can not be excluded that, in patients carrying the VUS mutations predicted to be pathogenic and the wild-type BRCA genes, alterations in other gene(s) may be responsible for BC risk. 

Author Response

The manuscript describes the possible association of two newly discovered MSH2 gene variants with BC risk. Authors used the pathogenic and non-pathogenic MSH2 mutations retrieved from several database to develop a classifier considering several physicochemical parameters. The classifier was then used to predict the pathogenicity of 4 VUS mutations identified in their cohort of BC patients. Overall, the manuscript is interesting and clearly written. However, some issues should be addressed.

Response: Thank you for your time.

 Abstract

Page 1, line 14: Did the Authors mean “kit” instead of “kid”?

Response: Thank you. Corrected.

Authors should correct MHS2 with MSH2. Authors should check this issue also throughout the text.

Response: Corrected.

I think that the last statement “...serve as new biomarker…” is too strong and overestimates the findings. To support Authors conclusions, an in vitro validation of mutation pathogenicity is needed and, moreover, the presence of these mutations should be demonstrated in a higher number of patients or in families with BC history. Conclusions need to be rewritten.

Response: Corrected as suggested by the reviewer.

Results and Discussion

Page 11, line 286: Authors stated that the p.Ala272Val variant is reported in ClinVar as a mutation with unknown clinical significance, but I found that it was classified as "Benign" on Oct 18, 2018. Authors should clarify this point.

Response: Indeed, the p.Ala272Val was re-classified in ClinVar on Oct 18, 2018 as pointed out by the reviewer. We missed this. However,  p.Ala272Val was classified in Greenwood Genetic Center study as VUS based on “American College of Medical Genetics (ACMG) Standards and Guidelines for the Interpretation of Sequence Variants”, as described in the method section of the paper. Lines 364-370 in the marked version.

Page 11, line 292: Authors should check the mutation reported for patient B4. Some lines above, Authors wrote that patient B4 carries the p.Met592Val mutation and not the p.Ala272Val.

Response: This was a typo. Corrected.

Page 11, line 296: Here and throughout the text Authors should change “BRACA” with “BRCA”.

Response: Thank you. Corrected.

Materials and Methods

Since its damaging effect on protein, it could be interesting a comment on the stop codon mutation that Authors found but that they excluded. Is it present in a patient without mutation in BRCA genes? Is there a family history of BC? What about the clinical features of this patient?

Response: The patient with stop codon MSH2:c.2785C>T also has MSH6: c.124C>T,c.3265T>C and BRCA2: c.4856A>G mutations. The patient has stage I Breast cancer and has a breast cancer family history. Thus, there are many factors, not only stop codon in MSH2 gene, contributing to disease. Text is added in the revision to point out that the patient has BRCA2 mutation.

Conclusions

Page 18, lines 481-482: Similarly to Abstract, the sentence should be rewritten. Authors should also point out that the impact of the two mutations should be confirmed in vitro and in a larger cohort of patients or in relatives with BC history. Otherwise, since Authors investigated only MSH2 gene, it can not be excluded that, in patients carrying the VUS mutations predicted to be pathogenic and the wild-type BRCA genes, alterations in other gene(s) may be responsible for BC risk.

Response: Conclusion is revised to reflect reviewer suggestion.

Round 2

Reviewer 2 Report

Authors solved all issues but on page 11, line 289 is still reported that in ClinVar the p.Ala272Val is classified as VUS. Authors should delete or correct the sentence.

This manuscript is a resubmission of an earlier submission. The following is a list of the peer review reports and author responses from that submission.

Round 1

Reviewer 1 Report

Wu et al. presented the manuscript indicated that two newly discovered variants in MSH2 gene, which codes for a 12 DNA mismatch repair (MMR) protein, can be associated with high risk of breast cancer. The conclusion suggests that two mutations can serve as new biomarkers for early detection of elevated for breast cancer. The study provides some insight to for the early diagnostics breast cancer. These studies were carried out carefully and have completed the writing describes the techniques and findings clearly. Although there are the abbreviations section, but suggest that the abbreviations should can be clear description in the text first appearance. 

Author Response

Thank you for your comments. As suggested, in the revision the abbreviations are described in the main text as well. 

Reviewer 2 Report

The authors tried to distinguish pathogenic from benign mutations for MHS2 genes variants via in silico analysis. They applied this analysis to identified four variants with unknown significance (VUS) in MHS2 gene. In consequence, p.Ala272Val and p.Met592Val of MSH2 gene, are predicted to be pathogenic mutations. They concluded that these two mutations are found in women with breast cancer who do not have mutations in BRCA1 and BRCA2 genes, and are able to use as new biomarkers for early detection of elevated breast cancer risk.

The MSH2 c.815C > T (p.Ala272Val) was previously described as causing partial exon skipping and it was identified in their work together with the path_BRCA2 c.9382C > T (p.R3128X) (Hereditary Cancer in Clinical Practice 2018).

Totally 138 alterations (previously described polymorphisms were not included) were reported, of which 58 affected MSH2 based on Danish person registry and the Danish parish registers. The MSH2 c.815C > T (Ala272Val) was indicated as reduced MSH2 function (Familial Cancer 2009).

In general, ‘pathogenic’ variants in disease genes related to phenotype means that functional relationship between phenotype and variants in vitro and in vivo has been identified with certainty. Then, this manuscript appears to be reporting significant discrimination between pathogenic and benign mutations for MHS2 genes variants based on in silico analysis, however, the impact is lost because of no functional analysis for p.Ala272Val and p.Met592Val of MSH2.

Author Response

Rev: The MSH2 c.815C > T (p.Ala272Val) was previously described as causing partial exon skipping and it was identified in their work together with the path_BRCA2 c.9382C > T (p.R3128X) (Hereditary Cancer in Clinical Practice 2018).

Response: In our data, the patients with p.Ala272Val and p.Met592Val in MSH2 do not have mutations in BRCA. This is an important distinction which implicates that p.Ala272Val and p.Met592Val in MSH2 alone are responsible to elevated breast cancer risk.

Rev: The MSH2 c.815C > T (Ala272Val) was indicated as reduced MSH2 function (Familial Cancer 2009).

Response: Although p.Ala272Val mutation is reported to reduce protein function, this mutation is still reported as a mutation with unknown clinical significance.

Rev: … the impact is lost because of no functional analysis for p.Ala272Val and p.Met592Val of MSH2.

Response: Unfortunately, we do not have access to appropriate experimental equipment to carry such investigations.

Round 2

Reviewer 2 Report

Rev: The MSH2 c.815C > T (p.Ala272Val) was previously described as causing partial exon skipping and it was identified in their work together with the path_BRCA2 c.9382C > T (p.R3128X) (Hereditary Cancer in Clinical Practice 2018).

Response 1: In our data, the patients with p.Ala272Val and p.Met592Val in MSH2 do not have mutations in BRCA. This is an important distinction which implicates that p.Ala272Val and p.Met592Val in MSH2 alone are responsible to elevated breast cancer risk.

2nd round Rev: In authors’ data, the patients with p.Ala272Val and p.Met592Val in MSH2 do not have mutations in BRCA. Please explain this discrepancy between the authors’ data and reported results as c.815C > T (p.Ala272Val) with BRCA2 c.9382C > T. Moreover, if the authors conclude that p.Ala272Val and p.Met592Val in MSH2 are responsible to elevated breast cancer ‘risk’, they have to analyze enough population sizes from clinical cases. If not, this is only speculation.

Rev 2/3: The MSH2 c.815C > T (Ala272Val) was indicated as reduced MSH2 function (Familial Cancer 2009). ・・・・・the impact is lost because of no functional analysis for p.Ala272Val and p.Met592Val of MSH2.

Response 2/3: Although p.Ala272Val mutation is reported to reduce protein function, this mutation is still reported as a mutation with unknown clinical significance. Unfortunately, we do not have access to appropriate experimental equipment to carry such investigations.

2nd round Rev: If there are no functional analysis for p.Ala272Val and p.Met592Val of MSH2, this is only speculation.